# Natural Yogurt Stabilized with Hydrocolloids from Butternut Squash (*Cucurbita moschata*) Seeds: Effect on Physicochemical, Rheological Properties and Sensory Perception

**Sergio A. Rojas-Torres, Somaris E. Quintana** and **Luis Alberto García-Zapateiro** *

Research Group of Complex Fluid Engineering and Food Rheology, University of Cartagena,
Cartagena 130015, Colombia; sergio1220rojas@gmail.com (S.A.R.-T.); squintanam@unicartagena.edu.co (S.E.Q.)
* Correspondence: lgarciaz@unicartagena.edu.co; Tel.: +57-675-2024

**Abstract:** Stabilizers are ingredients employed to improve the technological properties of products. The food industry and consumers have recently become interested in the development of natural ingredients. In this work, the effects of hydrocolloids from butternut squash (*Cucurbita moschata*) seeds (HBSS) as stabilizers on the physicochemical, rheological, and sensory properties of natural yogurt were examined. HBSS improved the yogurt's physical stability and physicochemical properties, decreasing syneresis and modifying the samples' rheological properties, improving the assessment of sensory characteristics. The samples presented shear thinning behavior characterized by a decrease in viscosity with the increase of the shear rate; nevertheless, the samples showed a two-step yield stress. HBSS is an alternative as a natural stabilizer for the development of microstructured products.

**Keywords:** butternut squash (*Cucurbita moschata*); seeds; pulp; hydrocolloid; rheology; sensorial

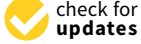



## 1. Introduction

Yogurt is a widely consumed dairy food that is internationally recognized for its health benefits, nutritional value, and digestibility [1,2]. Yogurts are produced by milk-controlled fermentation combining symbiotic cultures of *Streptococcus thermophilus* and *Lactobacillus delbrueckii spp. bulgaricus*, resulting in a product with creamy characteristics, a typical aroma, and a slightly acidic taste [3,4].

The most important attributes for consumer acceptance of yogurts are texture and firmness, related to viscosity and stability. The different ingredients employed for yogurt production such as nonfat dry milk, milk protein concentrate, and whey protein, improve texture, mouthfeel, appearance, viscosity, and consistency and prevent whey separation [5,6]. Moreover, the addition of phytochemicals, stabilizers, or other materials such as pectin, gelatin, inulin, or dietary fiber may enhance some sensory acceptance of yogurt and decrease syneresis [7]. Different hydrocolloids have been used as stabilizers for yogurt to improve their acceptance and extend their shelf life [8].

Hydrocolloids are polysaccharides and proteins utilized in industry to act as thickeners in relatively low concentrations due to their high molecular weight and technological functionality [9]. They are generally used as stabilizers due to their hydrophilic properties such as water retention, emulsion stability, and texture modification to improve texture or enhance properties such as appearance, mouthfeel, consistency, viscosity, and prevent whey separation temperature flocculation [6,10,11]. The rheological behavior of solution-added hydrocolloids is affected by the hydrocolloid backbone's structural features and its side chains, molecular weight, and conformation of the hydrocolloid molecules, as well as the solvent conditions [12].

Over the years, numerous papers on the natural and non-toxic stabilizer benefits for human health have been published [13]; recently, pectin-rich ingredients have been investigated to improve the gel characteristics, microstructure, texture, and rheology of

yogurt through interactions with a casein network (i.e., orange fiber, apple pomace, carrot cell) [14–16]. Plant seeds have good potential as a new hydrocolloid source regarding their safety, usability, and low production costs. These include carob, flaxseed, and white mustard as potential sources of hydrocolloids.

In the case of butternut squash (*Cucurbita moschata*) seeds, they are a source of proteins [17], polysaccharides [18], and bioactive components such as vitamins, provitamins, phytosterols, phospholipids, and fatty acids [19–21], also present technological properties such as solubility and emulsification capacity [22,23]. However, there are few reports on squash seed characterization and functional properties and their use in the development of food products. The present study was carried out to evaluate the effect of butternut squash seed hydrocolloids on the physicochemical, rheological, and sensory properties of yogurt.

## 2. Materials and Methods

### 2.1. Materials

Butternut squash was purchased at the local food market in the city of Cartagena (Colombia). Ethanol (99.5% purity) and hexane were obtained from Panreac (Barcelona, Spain). NaOH, acetic acid, and phenolphthalein were purchased from Sigma–Aldrich (St. Louis, MO, USA). Xanthan gum was purchased from Tecnas (Medellin, Colombia). Commercial pasteurized and homogenized milk and skim milk powder were purchased from a local Colombian market. Starter culture (*Lactobacillus bulgaricus* and *Streptococcus thermophilus)* strains were used in yogurt production. All other reagents were of analytical grade.

### 2.2. Methods

#### 2.2.1. Extraction of Hydrocolloids from Butternut Squash Seeds

Squash seeds were selected by color, washed with sodium hypochlorite 100 ppm for 5 min, and washed with distilled water; after that, the seeds were freeze-dried at −50 °C and 0.02 Pa for 8 h employing the Labconco Freezone 1.5 Liter Benchtop Freeze Dry equipment (Kansas City, MO, USA). Dried seeds were degreased using an oil extractor (Dulong model DL-ZYJ05) to obtain an oil-free material. Next, the degreased seeds were ground employing an MF 10 basic microfine grinder drive coupled with an MF 10.1 (cutting–grinding head, IKA, Germany) to reduce the particle size.

The extraction of hydrocolloids was done following the procedures described by Orgulloso-Bautista et al. [24] and Quintana et al. [25] with some modifications. The solubilization was carried out with distillate water at pH 10 for 4 h at 80 °C. The pH was adjusted employing acetic acid and NaOH. The mixture was separated by centrifugation for 15 min at 4000 rpm, and the supernatant was recollected. After that, the viscous solution was mixed with ethanol in a 1:1 ratio to precipitate the hydrocolloid-based extract and mixed for 2 h at $4.0 \pm 0.5$ °C. The mixture was centrifugated, and the precipitate was recollected and lyophilized during 48 h. Hydrocolloids from butternut squash (*Cucurbita moschata*) seeds were milled and stored at 5 °C until use.

#### 2.2.2. Production of Natural Yogurt

Different formulations of yogurt were prepared to employ different percentages of xanthan gum (XG) and hydrocolloids of butternut squash seed (HBSS) as stabilizers, as shown in Table 1. A control sample (M1) was prepared free of stabilizers; the other samples were prepared using 1% of a blend of XG and HBSS (M3, M4, and M5). Samples employing only XG or HBSS were prepared (M2 and M6, respectively).

**Table 1.** Experimental design of yogurt with the addition of hydrocolloids of butternut squash seed as stabilizers.

| Sample Code | Xanthan Gum mg/100 g | Hydrocolloids mg/100 g |
|:---:|:---:|:---:|
| M1 | 0 | 0 |
| M2 | 1.00 | 0 |
| M3 | 0.75 | 0.25 |
| M4 | 0.50 | 0.50 |
| M5 | 0.25 | 0.75 |
| M6 | 0 | 1.00 |

Yogurt was prepared according to Ladejvardi et al. [26] and Aziznia et al. [27] with some modifications. Fresh skim milk was supplemented with skim milk powder until reaching 16% soluble solids. The blend milk was heated at 90 °C for 5 min and rapidly cooled to 42 ± 1 °C to inoculate the lactic starters. The cultured milk was divided into 6 equal parts (120 mL) in sterile glass containers to develop the samples with different percentages of XG–HBSS blends; after that, all samples were incubated at 40 °C until a pH value of 4.5 was achieved. Each different batch of blends was agitated and stored at 5 °C until analysis. The formulations were manufactured in triplicate.

### 2.2.3. Physicochemical Properties and Proximal Composition of Yogurt

The physicochemical properties of yogurt were determinate after 24 h of processing following the method described by the Association of Official Analytical Chemistry [28]. The pH values were determined according to method 942.05/90 using a digital pH meter (Model HI 9124, Hanna Instruments, Woonsocket, RI, USA). Titratable acidity determination (TTA) was analyzed according to method number 967.21; 10 g of the samples were diluted into 250 mL using distilled water. Next, an aliquot of 50 mL was blended with 0.2 mL phenolphthalein as the indicator and was titrated with 0.1 N NaOH; TTA was expressed as mg of lactic acid per 100 g of the sample. Density was measured using a gravimetric method with grease pycnometers [29].

Moisture was determined by dehydration in an oven at 105 °C; ash was determined by incineration at 550 °C until constant weight. The fat determination was done using hexane as a solvent from Soxhlet extraction for 4 h. The total protein content was determined by Kjeldahl and total carbohydrates per difference [28].

### 2.2.4. Syneresis Measurement

The syneresis of yogurt was determined after 25 days of storage, following the procedures described by Säker and Rodriguez [30] with modifications. 10 g of yogurt sample were centrifuged at 5000 rpm for 20 min at 10 °C. After centrifugation, the clear supernatant was poured off, weighed, and used to determine the percentage (*w/w*) of syneresis (Equation (1)):

$$\% \text{ Syneresis} = \frac{\text{g supernatant whey}}{\text{g sample weight}} \times 100 \tag{1}$$

### 2.2.5. Color Parameters of Yogurt

The color parameters of lightness ($L^*$), red–green color ($a^* : + :$ red; $- :$ green), and yellow–blue color ($b^* : + :$ yellow; $- :$ blue) of the yogurt were measured by the colorimeter. The whiteness index ($WI$) and color variation ($\Delta E^*$) were calculated by Equations (2) and (3), respectively:

$$WI = 100 - \left[ (100 - L^*)^2 + a^{*2} + b^{*2} \right]^{0.5} \tag{2}$$

where the subscript *m* stands for the sample of yogurt formulated with XG and HBSS, and *c* stands for the control sample (M1).

### 2.2.6. Rheological Analysis

The rheological characterization of yogurt was carried out after 48 h of preparation in a controlled-stress rheometer (Modular Advanced Rheometer System Haake Mars 60, Thermo-Scientific, Dreieich, Germany) equipped with a coaxial cylinder (inner radius 11.60 mm, outer radius 12.54 mm, cylinder length 37.6 mm) based on Quintana et al. [31]. Each sample was held at the temperature of assay for 600 s to ensure the same thermal and mechanical history for each sample.

Viscous flow tests were carried out at a steady state, analyzing the variation of apparent viscosity (defined by the relationship between shear stress and shear rate) within a range of shear rates between 0.001 and 1000 $s^{-1}$ at 5, 10, 15, and 25 °C. Small-amplitude oscillatory shear (SAOS) tests were performed to obtain viscoelastic responses. In this way, stress sweeps were carried out at a frequency of 1 Hz, applying an ascending series of stress values from 0.001 to 1000 Pa at 5 °C to determine the linear viscoelasticity interval. Frequency sweeps were performed to obtain the mechanical spectrum using a stress value within the linear viscoelastic range in a frequency range between $10^{-2}$ and $10^{2}$ rad·$s^{-1}$. The data recorded included the storage modulus ($G'$), which provided the elastic component; the loss modulus ($G''$), which was related to the viscous components of the material; and the loss tangent (Tan $\delta$), which was the ratio $G''/G'$ and provided the ratio of elastic to the viscous response of the material under consideration.

### 2.2.7. Sensorial Analysis

Sensory evaluation was conducted by 30 tasters (15 males and 15 females, age 20–30 years, healthy subjects, lactose tolerant) using quantitative descriptive analysis. The panelists were instructed to evaluate each sample individually. Sensory analysis of the samples consisted of the evaluation of color, flavor, and consistency attributes employing the following hedonic scale descriptors: I dislike a lot = 1, I dislike a little = 2, I neither like nor dislike = 3, I like it = 4, and I like it a lot = 5. Equal amounts of each yogurt were prepared in 90 mL glass containers that were labeled, identified with three-digit random numbers, and equilibrated at room temperature for at least 1 h before consumption. The samples were served to panelists with a spoon and room-temperature water to cleanse the palate before presenting the samples, as described in technical guide GTC 165 [32]. The results were expressed as a percentage of acceptability.

### 2.2.8. Statistical Analysis

The data obtained were analyzed by ANOVA (unidirectional) using Statgraphics software (Centurion Version 16.1) to determine statistically significant differences ($p < 0.05$) between the samples submitted to the characterizations.

## 3. Results and Discussion

### 3.1. Physicochemical Properties of Yogurt

The physicochemical properties of the yogurt are shown in Table 2. The pH and acidity contributed to the organoleptic properties of the yogurt. During the incubation process, the pH reached values between 4.54 and 4.57, contributing to the characteristic odor and flavor [33] due to the breakdown of lactose by lactic acid bacteria [34]. A decrease in this value favored the contraction of protein clots formed by lactic bacteria, affecting the yogurt's sensory quality [35]. The acidity values (0.82 and 0.83% lactic acid) did not present significant variation ($p > 0.05$), preserving the flavor and texture of the yogurt with quality [36]. Next, the addition of XG and HBSS did not influence the density of yogurt ($p > 0.05$), presenting values between 1.06 and 1.08 g/mL that were within the normalized range [33].

**Table 2.** Physicochemical properties of yogurts with the addition of hydrocolloids of butternut squash seed as stabilizers.

| Sample Code | pH | Acidity % Acid Lactic | Density g/mL |
|---|---|---|---|
| M1 | 4.54 ± 0.01 [a] | 0.85 ± 0.005 [a] | 1.06 ± 0.00 [a] |
| M2 | 4.55 ± 0.01 [a] | 0.84 ± 0.005 [a] | 1.07 ± 0.00 [a] |
| M3 | 4.57 ± 0.01 [a] | 0.83 ± 0.005 [a] | 1.08 ± 0.00 [a] |
| M4 | 4.56 ± 0.05 [a] | 0.83 ± 0.005 [a] | 1.07 ± 0.00 [a] |
| M5 | 4.57 ± 0.01 [b] | 0.82 ± 0.005 [a] | 1.07 ± 0.00 [a] |
| M6 | 4.56 ± 0.05 [a] | 0.83 ± 0.005 [a] | 1.07 ± 0.00 [a] |

Results are expressed as mean ± standard deviation. Different letters in the same row express statistically significant differences ($p < 0.05$).

The syneresis in the product was a factor that required special attention due to the secretion of water on the gel structure. Syneresis can reflect a lower structural stabilization, becoming one of the worst defects in the final product, negatively affecting the consumer's acceptance and inferring the microbiological problem [37]. The syneresis of yogurt was evaluated during 25 days of storage ($5 \pm 0.1$ °C) (see Figure 1). The stabilizer-free sample (M1) presented the highest syneresis percentages compared to the others and showed an increase over time, followed by M2, M3, M4, M5, and M6.

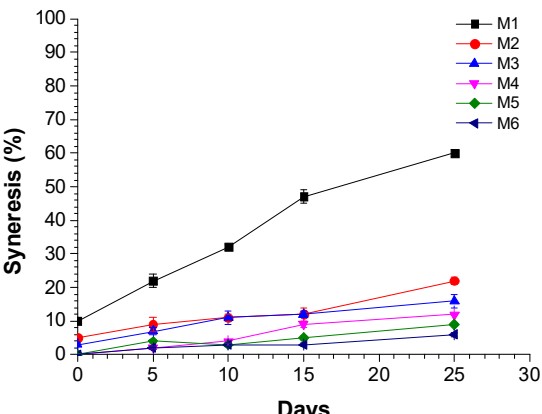

**Figure 1.** Syneresis (%) of yogurts with the addition of hydrocolloids from butternut squash seeds as stabilizers through 25 days of storage. Results represent average values ± SD (*n* = 3).

Samples with XG, HBSS, and their blends presented a significant decrease in their syneresis percentage ($p < 0.05$) in comparison with M1, indicating that stabilizers preserved phase separation, protecting the shelf life of yogurt [38]. The blend of XG with HBSS favored the water retention action and contributed to the mesh effect in the three-dimensional gel network formed in the yogurt, resulting in greater gel stability and decreased syneresis throughout the storage process [39]. In addition, sample M6, prepared only with HBSS, presented the lowest percentage of syneresis during the storage times; this could be explained by highly branched structure of HBSS, which could easily interact with other components and form bonds with the protein matrix, ions in the aqueous phase, and water. This reduction in syneresis occurred due to the effective immobilization of the aqueous phase by the gelatin in the yogurt, which decreased the susceptibility of the yogurt to this phenomenon.

*3.2. Color Analysis*

The optical properties of yogurts contributed to their sensory attributes and consumer acceptance. The color parameters of the samples are shown in Table 3. The highest luminosity values (L*) are attributed to their white color and bright. The samples presented significant differences ($p < 0.05$) in comparison with stabilizer-free yogurt (M1), showing

a linear increase with the addition of HBSS. The brightness of yogurt is related to fat globules and protein particle size, influencing the light reflectance of particles and scattering ability [40,41].

**Table 3.** Color parameters of yogurts with the addition of hydrocolloids of butternut squash seed as stabilizers. L*: luminosity; ΔE : change of color; WI: whiteness index.

| Sample Code | L* | ΔE | WI |
|---|---|---|---|
| M1 | 62.81 ± 1.94 [a] | – | 60.14 ± 0.36 [a] |
| M2 | 69.59 ± 0.40 [a] | 25.60 ± 2.63 [a] | 67.05 ± 0.36 [b] |
| M3 | 70.11 ± 0.44 [ab] | 58.20 ± 6.20 [b] | 70.81 ± 0.72 [ab] |
| M4 | 72.48 ± 1.25 [b] | 61.13 ± 7.75 [b] | 70.85 ± 0.93 [ab] |
| M5 | 74.07 ± 0.68 [b] | 79.15 ± 10.51 [c] | 71.16 ± 2.18 [b] |
| M6 | 76.11 ± 0.50 [b] | 93.36 ± 5.87 [d] | 73.88 ± 0.38 [b] |

Results are expressed as mean ± standard deviation. Different letters in the same row express statistically significant differences ($p < 0.05$).

ΔE represents the change in the color of the yogurt formulated, concerning the reference color (M1). The addition of XG and HBSS produced significant differences ($p < 0.05$), whereas M6 presented a difference of 75% from the control sample (M1). Similarly, the samples presented a linear increase of WI ($p < 0.05$) with the addition of the HBSS. Changes in the WI can be related to changes in the structure, resulting in different light scattering [42].

### 3.3. Rheological Properties

#### 3.3.1. Steady Shear Rate

The viscous flow of the formulated yogurts containing different concentrations of XG and BHSS was related to their intrinsic characteristics such as kinetic stability and mean particle size. The samples presented a heterogeneous, three-dimensional network composed of continuously connected casein strands and whey protein aggregates [43,44]. All samples presented a nonlinear viscosity–shear rate relation (Figures A1 and A2), showing a decrease of apparent viscosity with the increase in shear rate, indicating a typical, non-Newtonian flow behavior-type shear thinning. This behavior was possibly due to the aligned, in-direction flow of stiff polymer molecules, decreasing the interaction between adjacent polymer molecule chains [45,46] and disrupting the flocculated casein micelle network while shear force was applied [47]. This viscous flow behavior can be observed in samples M2 and M6 in Figure 2a.

Nevertheless, the curve inflection points suggested apparent yield stress [48] in all cases. In the same way, yogurt with the addition of HBSS as a stabilizer showed decreased apparent yield stress in all batches, related to higher degrees of crosslinking of milk proteins in acidified milk [48]. The results suggested that shear (approximately $10 \text{ s}^{-1}$) affected the yogurt structure.

Figure 2b shows the full viscosity–shear stress curves of samples M2 and M6 (Appendix A). All samples presented a two-step yielding point from the decline in viscosity. The extrapolation approach reported by Zhu et al. [49] was employed to obtain and compare the yield stress values of samples. The two stress values were associated with a static yield stress $\tau_{o-1}$ (first decline) and fluidic yield stress, $\tau_{0-2}$ (second decline). The same behavior has been reported in the literature, e.g., in yogurts formulated with monk fruit extract as a sweetener [50], yogurt containing selecan [51], and fat-free stirred yogurt [44], when the authors associated this behavior with the apparent yield stress [50] that could be due to breakage of the yogurt structure and the sensitive reaction of yogurts to shear stress due to its quasi-stable status [51]. However, there is no evidence of any analysis reports of these two areas regarding this behavior.

Table 4 shows the low shear viscosity ($\eta_0$) and yield stress ($\tau_o$) parameters in the static and fluidic yield stress of all samples at different temperatures.

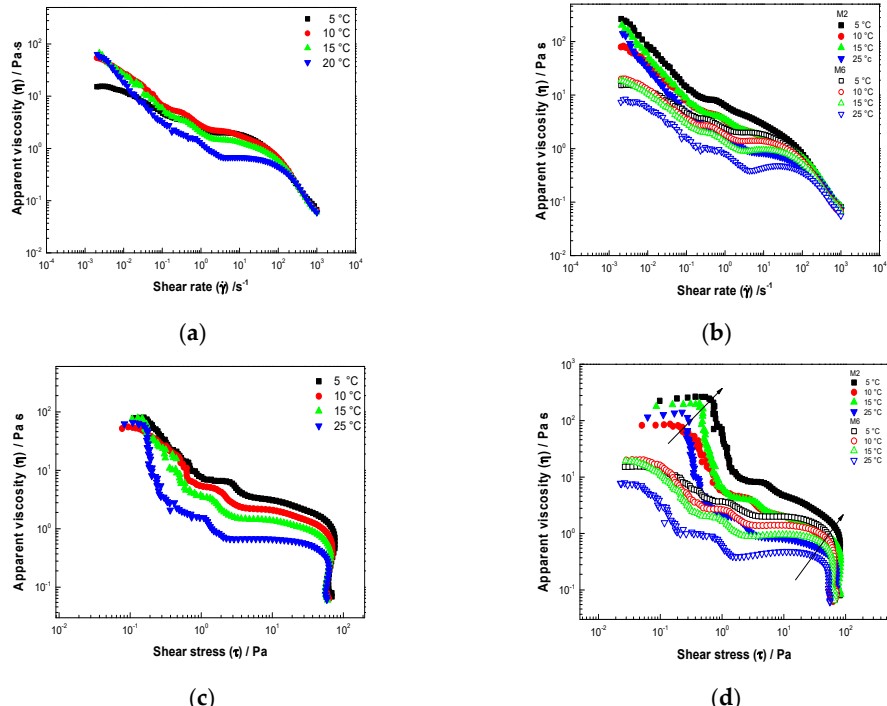

**Figure 2.** Flow curves showing apparent viscosity as functions of the shear rate of samples M1 (**a**) and M2 and M6 (**b**) and the shear stress of M1 (**c**) and M2 and M6 (**d**) of yogurts with the addition of hydrocolloids from butternut squash seeds as stabilizers.

**Table 4.** Low shear viscosity ($\eta_0$) and yield stress ($\tau_o$) parameters of yogurts with the addition of hydrocolloids of butternut squash seeds as stabilizers at different temperatures.

| Sample Code | Temp. °C | $\eta_{0\,(First)}$ Pa·s | $\eta_{0\,(Second)}$ Pa·s | $\tau_{o-1}$ Pa | $\tau_{0-2}$ Pa |
|---|---|---|---|---|---|
| M1 | 5 | 79.20 [a] | 3.09 * | 0.15 [b] | 26.52 [b] |
| | 10 | 51.12 [a] | 2.13 + | 0.14 [b] | 30.44 [b] |
| | 15 | 76.78 [a] | 1.48 + | 0.13 [b] | 28.93 [b] |
| | 25 | 66.20 [a] | 0.67 | 0.13 [b] | 29.72 [b] |
| M2 | 5 | 256.3 [b] | 2.99 * | 0.66 [a] | 55.61 [b] |
| | 10 | 84.83 [b] | 1.96 + | 0.23 [a] | 22.14 [b] |
| | 15 | 193.70 [b] | 1.68 +· | 0.37 [a] | 62.35 [b] |
| | 25 | 126.78 [b] | 0.87 + | 0.28 [a] | 40.96 [b] |
| M3 | 5 | 299.50 [c] | 8.10 * | 0.61 [a] | 24.25 [d] |
| | 10 | 211.04 [c] | 3.50 + | 0.43 [a] | 20.07 [d] |
| | 15 | 265.04 [c] | 3.06 +· | 0.52 [a] | 19.28 [d] |
| | 25 | 183.84 [c] | 1.30 | 0.28 [a] | 22.32 [d] |
| M4 | 5 | 93.93 [d] | 8.15 * | 0.15 [ab] | 23.32 [d] |
| | 10 | 220.07 [d] | 4.28 + | 0.48 [ab] | 24.09 [d] |
| | 15 | 90.97 [d] | 1.92 + | 0.14 [ab] | 23.06 [d] |
| | 25 | 66.28 [d] | 0.80 + | 0.15 [ab] | 23.03 [d] |
| M5 | 5 | 37.52 [e] | 3.04 * | 0.12 [b] | 35.82 [c] |
| | 10 | 30.75 [e] | 2.05 + | 0.12 [b] | 27.57 [b] |
| | 15 | 28.01 [e] | 1.33 + | 0.08 [b] | 19.42 [b] |
| | 25 | 29.30 [e] | 2.06 | 0.12 [b] | 26.87 [b] |
| M6 | 5 | 14.40 [f] | 8.65 * | 0.05 [ab] | 27.69 [b] |
| | 10 | 20.01 [f] | 1.42 + | 0.07 [ab] | 19.75 [c] |
| | 15 | 18.92 [f] | 1.38 + | 0.91 [ab] | 12.51 [c] |
| | 25 | 7.20 [f] | 0.05 | 0.05 [ab] | 11.07 [c] |

Different letters in the same columns express statistically significant differences ($p < 0.05$) for temperature, and different symbols express statistically significant differences ($p < 0.05$) for hydrocolloids.

From $\eta_0$, sample M1 (stabilizer-free) presented the lowest value. $\eta_{0\ (First)}$ did not show significant differences ($p > 0.05$) with the temperature, but it presented variation with the percentage of gums ($p < 0.05$). The samples prepared with XG (M2) showed the highest $\eta_0$, i.e., the highest first-Newtonian region, followed by samples prepared with XG + BHSS, although the decrease of XG showed a reduction of $\eta_0$. This high viscosity was attributed to dominating Brownian Forces at low shear rates that randomized the particles and formed many doublets and aggregates [52]. $\eta_{0\ (second)}$ decreased with the temperature ($p < 0.05$) and did not present significative differences with the percentage of hydrocolloids.

For the yield stress values, $\tau_{o-1}$ and $\tau_{0-2}$ did not present significant differences ($p > 0.05$) with the temperature but varied with the percentages of hydrocolloids ($p < 0.05$). In addition, $\tau_{o-1}$ in yogurt was linked to the breakdown of the interconnected network of flocs/aggregates. In contrast, the further collapse of flocs/aggregates into smaller flocs or individual particles was associated with $\tau_{0-2}$ [53]. Based on these designs and rheological results, the use of HBSS as an additive for the formulation of microstructure food matrices can be employed alone or blended with other hydrocolloids.

The temperature dependence of apparent viscosity at a specific temperature was evaluated using an Arrhenius-type equation:

$$\eta = \eta_i \cdot \left( \frac{E_a}{RT} \cdot \right) \tag{3}$$

where $\eta$ is the viscosity at 10 s$^{-1}$ ($\eta_{\dot{\gamma}=10s^{-1}}$), $\eta_i$ is a constant describing the viscosity coefficient at a reference temperature (Pa·s), $E_a$ is the activation energy (J·mol$^{-1}$), R is the gas constant (8.3144 J·mol$^{-1}$·K$^{-1}$), and T (K) the temperature [54]. The activation energy ($E_a$) is an important parameter of kinetic studies. The calculation was obtained by plotting the ln $(\eta_{\dot{\gamma}=10s^{-1}})$ against 1/T (Figure 3).

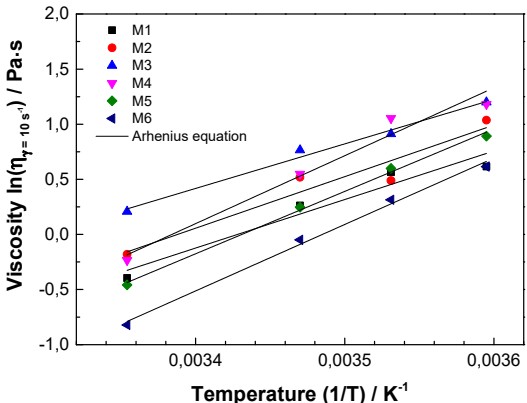

**Figure 3.** Viscosity at 10 s$^{-1}$ ($\eta_{\dot{\gamma}=10s^{-1}}$) versus 1/temperatures fitted to the Arrhenius equations of yogurt samples with additions of hydrocolloids of butternut squash seeds as stabilizers.

According to the linear fitting slope, the parameters adjusted to the Arrhenius equation are shown in Table 5. The *Ea* of yogurts were 485.09 and 734.03 J/mol, which had a high degree of fit ($R^2 > 0.92$). In this case, the sign inside the exponential function was positive because the viscosity decreased when the temperature increased [55]. The addition of stabilizers increased *Ea* values, showing a slight increase with the addition of HBSS. Therefore, the activation energy values indicated that the viscosity of the higher amounts of HBBS in the yogurt samples exhibited higher temperature dependence.

**Table 5.** Arrhenius equation parameters for yogurts with the addition of hydrocolloids of butternut squash seeds as stabilizers.

| Sample Code | $\eta_i$ Pa·s | Ea J/mol | $R^2$ |
|---|---|---|---|
| M1 | $0.063 \times 10^{-6}$ [a] | 528.55 [a] | 0.98 |
| M2 | $3.728 \times 10^{-11}$ [a] | 561.05 [b] | 0.92 |
| M3 | $2.314 \times 10^{-12}$ [a] | 485.09 [c] | 0.98 |
| M4 | $4.55 \times 10^{-16}$ [a] | 734.03 [d] | 0.98 |
| M5 | $2.60 \times 10^{-11}$ [a] | 679.55 [e] | 0.98 |
| M6 | $5.24 \times 10^{-9}$ [a] | 723.26 [f] | 0.99 |

Results are expressed as mean $\pm$ standard deviation. Different letters in the same column express statistically significant differences ($p < 0.05$).

### 3.3.2. Stress Sweep

The stress sweep was done in all samples to determine the viscoelastic linear region (VLR), evaluating the behavior of the storage ($G'$) and loss moduli ($G''$). It was observed that in all cases, under low-stress conditions, the modulus $G'$ and $G''$ were linear, and when the stress was sufficiently large, there was a substantial decrease, showing the slope tended to zero. At this point, the linearity profile deviated; it was considered the critical stress considered as the static yield stress $\tau_c$. Next, a second decrease was observed; nevertheless, the stress kept increasing, and it was difficult to determine the dynamic yield stress. Thus, the obtained curves corroborated the two-yield point's presence in all cases, recognized by the slope's change in the stress sweep curve (Figure 4).

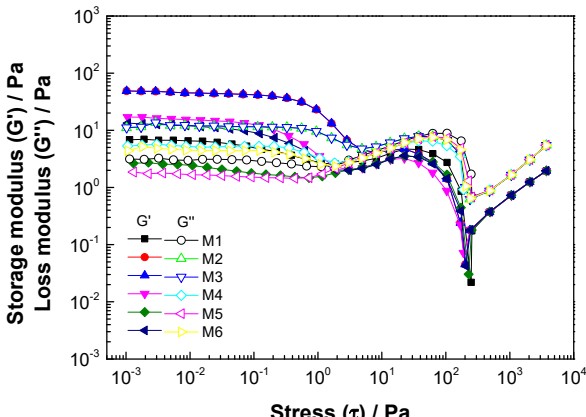

**Figure 4.** Stress sweep of yogurt samples with the addition of hydrocolloids of butternut squash seed as stabilizers at 5 °C.

For all samples, different linear and nonlinear viscoelastic regions could be observed. As shown in Table 6, in the linear viscoelastic regions, $G'$ were higher than $G''$; remaining constant indicated that the yogurt behaved like a viscoelastic solid. The samples prepared without stabilizer (M1) and with XG or HBSS alone (M2 and M6) did not present significative differences ($p < 0.05$) in $\tau_c$, and the interaction of XG–HBSS (M3 and M4) presented higher $\tau_c$, values, rationed with the crosslinking of the molecules.

**Table 6.** Critical stress $\tau_c$ and elastic $G'$ and viscous $G''$ moduli of yogurt samples with the addition of hydrocolloids of butternut squash seeds as stabilizers at 5 °C.

| Sample Code | $\tau_c$ Pa | $G'$ Pa | $G'$ Pa | Crossover $G' = G''$ Pa |
|---|---|---|---|---|
| M1 | 0.15 [a] | 6.359 [a] | 3.07 [a] | 2.57 [cd] |
| M2 | 0.15 [a] | 45.34 [b] | 11.60 [b] | 4.92 [b] |
| M3 | 0.25 [ab] | 53.74 [c] | 14.10 [c] | 6.49 [a] |
| M4 | 0.14 [a] | 15.39 [d] | 5.28 [d] | 3.09 [c] |
| M5 | 0.53 [b] | 2.152 [e] | 1.64 [e] | 1.47 [a] |
| M6 | 0.15 [a] | 11.65 [f] | 4.53 [f] | 2.48 [d] |

Results are expressed as mean $\pm$ standard deviation. Different letters in the same column express statistically significant differences ($p < 0.05$).

### 3.3.3. Frequency Sweep

The structural stability of the storage ($G'$) and loss moduli ($G''$), the complex modulus ($G^*$), and the loss tangent (tan $\delta$) are the main rheological parameters reported for yogurts [56–58]. The frequency sweeps of yogurt showed than the storage moduli ($G'$) were higher than the loss moduli ($G''$) over the entire range of frequencies studied, indicating that the elastic components were greater than the viscous, as shown in Figure 5.

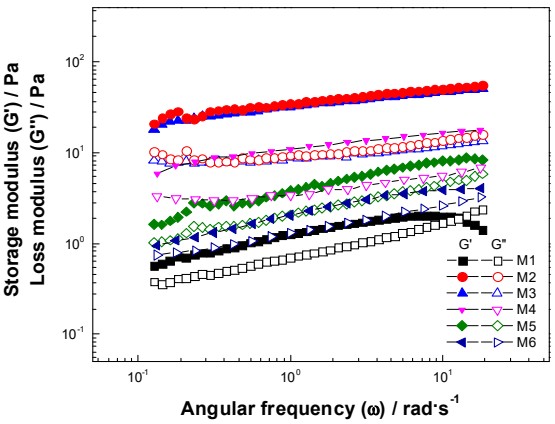

**Figure 5.** Frequency sweep of yogurts with additions of hydrocolloids of butternut squash seeds as stabilizers at 5 °C.

The solid-like properties ($G' > G''$) were observed to be dominant in XG + HBSS yogurt. Higher $G'$ and no crossover with $G''$ were documented as strong gel-like behavior, attributed to the interaction of casein with stabilizers because electrostatic bond formation at a low pH might have led to dense packing of the protein gel structure with hydrocolloids [56,59]. Values of $G'$ at $\omega = 1$ rad·s$^{-1}$ (Table 7) were used to compare the effects of stabilizers on the samples, presenting a decrease with the addition of HBSS ($p < 0.05$), while $G'$ was in the following order: M2 $\geq$ M3 $\geq$ M5 > M6. Nevertheless, the stabilizer-free sample presented crosslinking at high frequencies, showing the weakest behavior, which did not occur with the samples that contained XG and HBSS, and the stabilizer free sample could lose the desired characteristics by the consumer when the set of internal macromolecular networks of the yogurt lost the ability to trap internal fluids [60]. Moreover, the sample with XG (M2) presented a high $G'$ in the same way of the samples with 0.25 and 0.5 g/100 g of HBSS (M3 and M4, respectively), explained by the ability of XG to interact with the milk protein, altering the gel network [16]. The samples prepared only with HBSS (M6) presented intermediate values of $G'$ and $G''$ with gel-like behavior. The differences seen in the rheological behavior may have been due to the creation of protein–casein complexes. Similar values were obtained by Kumar and Sasmai [61] in the development of milk gel with the addition of *Cucurbita* seed extract.

**Table 7.** Values for the viscoelastic parameters of $G'$ (storage modulus) and $G''$ (loss modulus) in the frequency sweep of yogurt samples at 5 °C at 1 rad·s$^{-1}$.

| Sample Code | $G'$ Pa | $G''$ Pa | Tan $\delta$ |
|:---:|:---:|:---:|:---:|
| M1 | 1.25 [a] | 0.70 [a] | 0.56 [a] |
| M2 | 32.04 [b] | 8.97 [a] | 0.28 [b] |
| M3 | 34.35 [c] | 9.33 [a] | 0.28 [b] |
| M4 | 3.80 [d] | 2.08 [a] | 0.55 [a] |
| M5 | 10.88 [e] | 3.35 [a] | 0.30 [b] |
| M6 | 2.05 [f] | 1.30 [a] | 0.63 [a] |

Different letters in the same column express statistically significant differences ($p < 0.05$).

The obtained values of the loss tangent (Tan $\delta = G''/G'$) were between 0.27 and 0.63, indicating a wide range of behavior between a weak gel and a strong elastic gel. The addition of XG decreased the Tan $\delta$ associated with the formation of a stronger gel; next, the addition of HBSS (sample with XG + HBSS) indicated a modification of the gel structure. Samples with XB+HBSS presented similar behavior, and then the employee of HBSS (M6) increased the viscous behavior associated with the weaker gel structure. The addition of HBSS is an alternative for developing food products to increase their viscosity or thickness and improve the water binding ability and texture of yogurt. The choice of hydrocolloid depends on the attributes desired in the final product and the conditions of processing.

*3.4. Sensory Analysis*

Sensory evaluation of food products is a quality index of organoleptic properties such as texture, flavor, and appearance that is vital for the overall acceptability by consumers. The concentration of hydrocolloids plays an important role in governing yogurt's stability and texture and finds a correlation with improved quality and sensory perception [57,58]. Figure 6 shows the average results of the evaluated parameters.

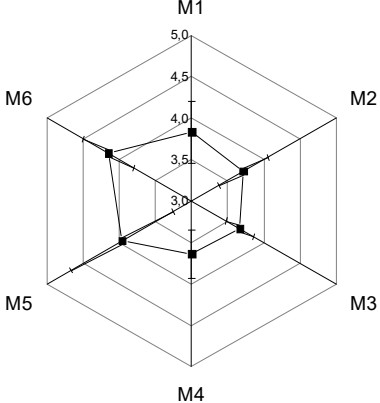

**Figure 6.** Global quality of yogurts with additions of hydrocolloids from butternut squash seed as stabilizers.

The samples prepared only with HBSS (M6) showed the highest average values for all assessed items, which corresponded to "I like it a lot", while the samples prepared with XG and blended with hydrocolloids corresponded to "I neither like nor dislike". Global quality involved all attributes. The color characteristic showed statistical differences ($p < 0.05$) between control samples with XG + HBSS, which resulted in greater scattering of light, and therefore, the products contained less luminosity and whiteness [62]. Nevertheless, similar results were obtained with the control samples and yogurt prepared with HBSS, produced by the scattering of light by the milk constituents: fat globules, casein micelles, colloidal calcium phosphate, some pigments, and riboflavin [63]. This sensory parameter's importance lies in the fact that it is one of the leading causes of a dairy product being

bought by the consumer or rejected. However, it does not reflect the same taste or nutritional value [62]. Similarly, the sample with HBSS presented the highest odor acceptance (*p* < 0.05) (Table 8).

**Table 8.** Sensorial analysis of yogurts with the addition of hydrocolloids of butternut squash seeds as stabilizers.

| Parameters | Color | Odor | Flavor | Consistency |
|---|---|---|---|---|
| M1 | 4.10 ± 0.71 [ab] | 3.70 ± 0.70 [ab] | 4.16 ± 0.59 [a] | 3.36 ± 1.32 [a] |
| M2 | 4.00 ± 0.72 [a] | 3.76 ± 0.77 [ab] | 3.90 ± 0.99 [a] | 3.23 ± 1.07 [a] |
| M3 | 3.83 ± 0.87 [a] | 3.56 ± 0.72 [a] | 3.86 ± 1.10 [a] | 3.46 ± 0.93 [a] |
| M4 | 3.86 ± 0.86 [a] | 3.56 ± 0.81 [a] | 3.90 ± 0.92 [a] | 3.26 ± 0.98 [a] |
| M5 | 4.16 ± 0.74 [ab] | 3.96 ± 0.55 [bc] | 3.96 ± 0.96 [a] | 3.76 ± 0.93 [a] |
| M6 | 4.45 ± 0.72 [ab] | 4.26 ± 0.78 [c] | 4.26 ± 0.86 [a] | 3.63 ± 1.29 [a] |

Results are expressed as mean ± standard deviation. Different letters in the same column express statistically significant differences (*p* < 0.05).

According to the panelists, the addition of blends of hydrocolloids (XG+HBSS) did not significantly affect the consistency (*p* > 0.05). The sensorial parameters such as creaminess, thickness, consistency, smoothness, and taste perception were related strongly to the rheological properties. The addition of hydrocolloids modified the rheology and improved the physical stability and overall mouthfeel properties to increase desirable general characteristics [64]. The samples with a moderate viscosity and storage and loss moduli presented the highest values of the evaluation of sensorial parameters. Considering the sensory analysis results altogether, it could be concluded that the application of HBSS as a food additive could be possible, with positive effects on the texture and sensory attributes and without the consequent appearance of unwanted effects.

*3.5. Proximal Composition of a Selected Standardized Product*

Table 9 shows the data from the proximal composition of M6, considered the best sample due to its attributes upon sensory evaluation carried out in comparison with the control sample. The proximal composition results indicated that the selected yogurt (M6) presented carbohydrate values of 8.18 ± 0.43%, a moisture value of 79.58 ± 0.47%, a fat value of 4.68 ± 0.11%, a fiber value of 0.89 ± 0.02%, a protein value of 6.10 ± 0.18%, and an ash value of 0.56 ± 0.05%. The data were compared with common and probiotic yogurts and presented higher carbohydrate and protein contents related to the content of HBSS due to the high content of carbohydrate and proteins in butternut squash seeds [17].

**Table 9.** Proximal compositions of yogurt with additions of hydrocolloids from butternut squash seeds (M6), common, and probiotic yogurt.

| Parameters | M6 | Common Yogurt * | Probiotic Yogurt * |
|---|---|---|---|
| Moisture (%) | 79.58 ± 0.47 | 88.72 | 88.74 |
| Ash (%) | 0.56 ± 0.05 | 0.72 | 0.71 |
| Fat (%) | 4.68 ± 0.11 | 2.45 | 2.42 |
| Carbohydrates (%) | 8.18 ± 0.43 | 5.24 | 5.20 |
| Proteins (%) | 6.10 ± 0.18 | 2.90 | 2.92 |
| Fiber (%) | 0.89 ± 0.02 | – | – |

* Source: [65].

## 4. Conclusions

The hydrocolloids from butternut squash (*Cucurbita moschata*) seeds were evaluated as stabilizers in yogurt samples to preserve the physicochemical properties, flavor, and texture of samples as well as their activity on yogurt starter bacteria. Yogurts present a non-Newtonian flow behavior-type shear thinning, with inflection points in the curves (approximately $10 \text{ s}^{-1}$) related to the presence of a yield point. The evaluation of stress in the function of the shear rate showed a two-step yielding point: static yield stress (first decline) and fluidic yield stress (second decline) associated with the breakdown of the interconnected network and the further collapse of flocs/aggregates into smaller flocs or individual particles; in addition, the viscoelastic properties of the yogurts presented solid-like properties in a typical strong gel-like behavior. HBSS contributed to the increase in the desirable overall characteristics that presented the highest valorization related to the highest WI and acceptance values for panelists. The addition of HBSS is an alternative for increasing the viscosity or thickness of food products and improving the water binding ability and texture of yogurt. The choice of hydrocolloid depends on the attributes desired in the final product and the conditions of processing. The employment of different hydrocolloids affects product yield, texture, mouthfeel, sensory properties, and consumer acceptance.

**Author Contributions:** Conceptualization, S.A.R.-T., S.E.Q. and L.A.G.-Z.; Methodology, S.A.R.-T. and L.A.G.-Z.; Software, S.E.Q.; Validation, S.E.Q. and L.A.G.-Z.; Formal Analysis, S.A.R.-T., S.E.Q. and L.A.G.-Z.; Investigation, S.A.R.-T. and L.A.G.-Z.; Resources, L.A.G.-Z.; Writing—Original Draft Preparation, S.A.R.-T., S.E.Q. and L.A.G.-Z.; Writing—Review and Editing, S.E.Q. and L.A.G.-Z.; Supervision, L.A.G.-Z.; Project Administration, L.A.G.-Z.; Funding Acquisition, S.E.Q. and L.A.G.-Z. All authors have read and agreed to the published version of the manuscript.

**Funding:** This research was funded by the Ministry of Science, Technology, and Innovation—MinCiencias (Contract 368-2019, Research Project No. 110780864755).

**Institutional Review Board Statement:** Not applicable.

**Informed Consent Statement:** Not applicable.

**Conflicts of Interest:** The authors declare no conflict of interest.

## Appendix A

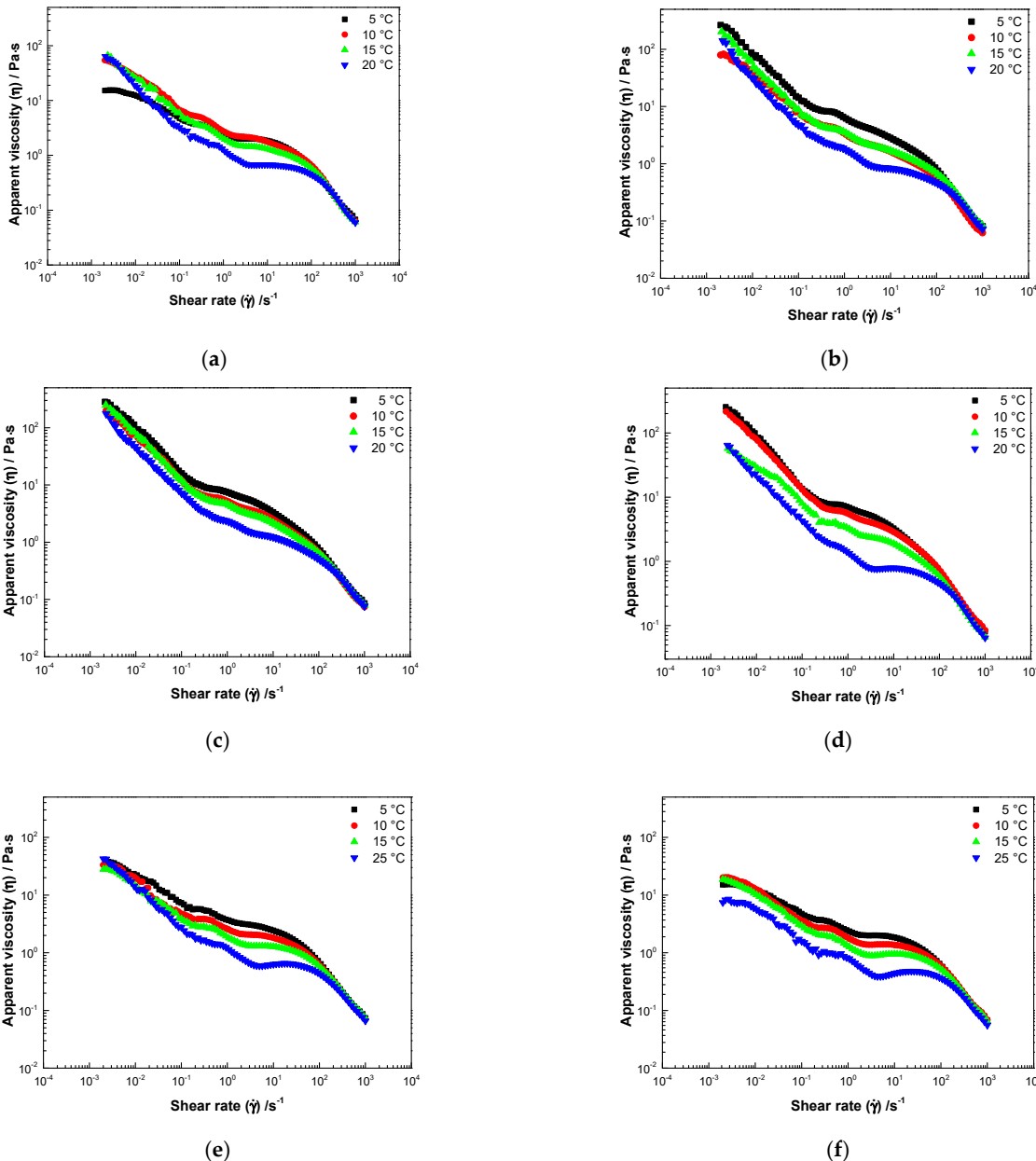

**Figure A1.** Flow curves show apparent viscosity in the shear rate of yogurts with the addition of hydrocolloids of butternut squash seeds as stabilizers: (**a**) M1, (**b**) M2, (**c**) M3, (**d**) M4, (**e**) M5, (**f**) M6.

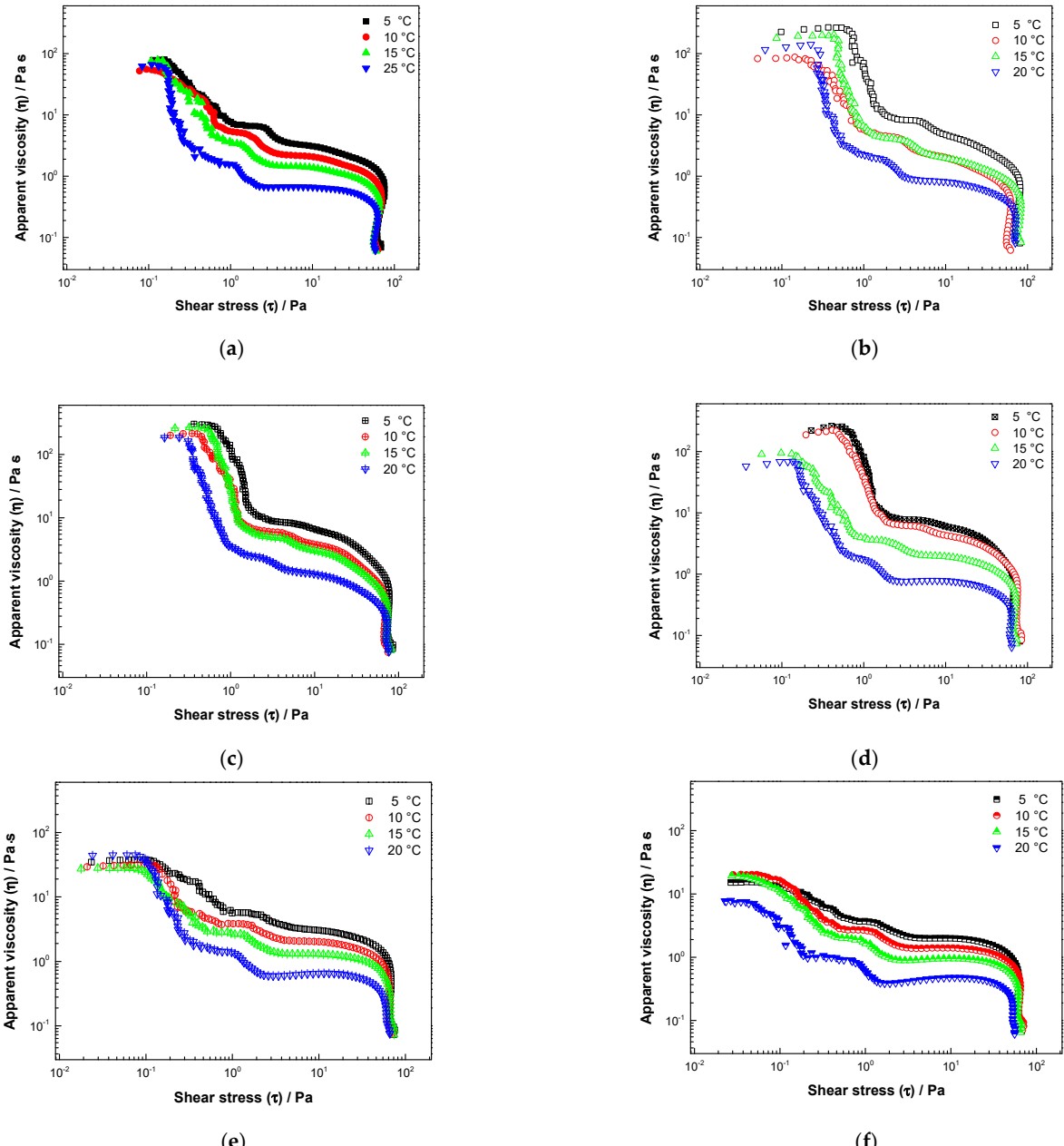

**Figure A2.** Flow curves showing apparent viscosity in functions of shear stress of yogurts with the addition of hydrocolloids of butternut squash seeds as stabilizers (**a**) M1, (**b**) M2, (**c**) M3, (**d**) M4, (**e**) M5, (**f**) M6.

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
