# Peer review of "Natural Yogurt Stabilized with Hydrocolloids from Butternut Squash (Cucurbita moschata) Seeds: Effect on Physicochemical, Rheological Properties and Sensory Perception"

_fluids, doi:10.3390/fluids6070251_

Round 1
Reviewer 1 Report
The manuscript entitled "Natural yogurt stabilized with hydrocolloids from butternut squash (Cucurbita moschata) seeds: effect on physicochemical, rheology properties and sensory perception" is a significant study with impact on current concerns at international level in food industry.
This article is well designed and demonstrates a systematic scientific approach and explanation of the observations. The Introduction section is well organized. Also, the following sections need not any re-organization, are well written with satisfactory scientific explanations, references and correlations. Results are clearly described and analysed with good quality of pictures. Additionally, the obtained results are innovative and relevant in relation on the subject. Conclusions are well written, providing useful information.
I have some minor concerns before I can recommend it for publication.
- The manuscript is very well written nevertheless I found some minor writings errors (lines 152, 236: viscous flow should be rewrite: viscoelastic flow, 273, Table 5: Ea, 303: linear,…)
- Syneresis of yogurts with the addition of hydrocolloids was measured in the period of 25 days. On which day the rheological properties of yogurt were measured, this information should be added into Material and Methods section. Surely the properties will change over time, do you plan to measure them?
- I do not understand the sentence (line 338) “Values of G´…was in the following order M1>M2…. The M1 has the smallest value of G´, is it mistake?
- It would be better for the reader if the authors add values of phase angle into the Table 7. From phase angle values can be see the elasticity of the yogurts with hydrocolloid.
Based on the above concerns, I suggest a minor revision.
Author Response
1. The manuscript is very well written nevertheless I found some minor writings errors (lines 152, 236: viscous flow should be rewrite: viscoelastic flow, 273, Table 5: Ea, 303: linear,…)
RE: The phrase Viscous Flow is correct. In this section, the test method is explained and related to a viscosity test in steady-state (dynamic viscosity). The term viscoelastic is related small-amplitude oscillatory shear (SAOS) tests.
2. Syneresis of yogurts with the addition of hydrocolloids was measured in the period of 25 days. On which day the rheological properties of yogurt were measured, this information should be added into Material and Methods section. Surely the properties will change over time, do you plan to measure them?
RE: The rheological properties were evaluated at 48 h of preparation. The studies on the rheological properties over the time are considering to other studies.
The text was modified:
L137: “The rheological characterization of yogurt was carried out after 48 h of preparation in a controlled-stress rheometer (Modular Advanced Rheometer System Haake Mars 60, Thermo-Scientific, Germany), based on Quintana et al., [31] equipped with a coaxial cylinder (inner radius 11.60 mm, outer radius 12.54 mm, cylinder length 37.6 mm). Each sample was equilibrated 600 seconds before the rheological test to ensure the same thermal and mechanical history for each sample”.
3. I do not understand the sentence (line 338) “Values of G´…was in the following order M1>M2…. The M1 has the smallest value of G´, is it mistake?
RE: Yes, it´s a mistake, the correct order is M2 ≥ M3 ≥ M5 > M6
4. It would be better for the reader if the authors add values of phase angle into the Table 7. From phase angle values can be see the elasticity of the yogurts with hydrocolloid.
RE: The loss tangent values were added on Table 7.
The following text was added:
L356: “The obtained values of loss tangent (Tan δ (=G^'')⁄(G^')) were between 0.27 and 0.63, indicating a wide range of behavior between and a weak gel, a strong elastic gel. The addition of XG decrease the Tan δ associate with the formation of stronger gel; then the addition of HBSS (sample with XG+HBSS) indicate a modification of gel structure; samples with XB+HBSS present similar behavior, then the employee of HBSS (M6) increase de viscous behavior associate with the weaker gel structure. The addition of HBSS is an alternative to develop the product to increase the viscosity or thickness of food products and improve the water binding ability and texture of yogurt. The choice of hydrocolloid depends on the attributes desired in the final product and the condition of processing.”
Reviewer 2 Report
The authors presented an interesting study featuring the effect of hydrocolloids seeds on both the rheological properties and the sensory perception of the resulting yogurt. The main conclusion is that the addition of hydrocolloids lead to a more stable as well as to a more-liked yogurt.
The manuscript is written clearly and the conclusions are supported by the data. I have just few minor comments I would like to see addressed before recommending the manuscript for publication.
- On page 4, when describing the rheometer, it seems that the inner radius is larger than the outer radius.
- Can the authors clarify the meaning of “apparent viscosity” in comparison to the standard viscosity? This will help readers less familiar with the concept.
- Please enlarge the font size, legend size and symbol size for all the figures.
- I think it would be good to move the data for M1 available in the supplementary information in the main manuscript in Figure 2. Being that a control, it will be easy to appreciate the effect of hydrocolloids on yield stress and flow curve.
- With reference to Figure 2, are all the measured values reliable in terms of torque? I am just mindful of the fact that the torque of some points at low shear may be smaller than the resolution of the instrument.
- Can the authors clarify the origin of the minimum observed in Figure 4?
- On Page 11, it is written that G’ of M1 is larger than M2. However, from the graph it seems that M1<M2.
Author Response
1. On page 4, when describing the rheometer, it seems that the inner radius is larger than the outer radius.
RE: The test was corrected:
L137: “The rheological characterization of yogurt was carried out after 48 h of preparation in a controlled-stress rheometer (Modular Advanced Rheometer System Haake Mars 60, Thermo-Scientific, Germany), based on Quintana et al., [31] equipped with a coaxial cylinder (inner radius 11.60 mm, outer radius 12.54 mm, cylinder length 37.6 mm).”
2. Can the authors clarify the meaning of “apparent viscosity” in comparison to the standard viscosity? This will help readers less familiar with the concept.
RE: The test was improved.
L143: “Viscous flow tests were carried out at a steady-state, analyzing the variation of apparent viscosity (defined by the relationship between shear stress and shear rate) in a range of shear rates between 0.001 and 1000 s-1 at 5, 10, 15, and 25 °C.”
3. Please enlarge the font size, legend size and symbol size for all the figures.
RE: The size of all figures was modified.
4. I think it would be good to move the data for M1 available in the supplementary information in the main manuscript in Figure 2. Being that a control, it will be easy to appreciate the effect of hydrocolloids on yield stress and flow curve.
RE: The data of M1 was added to Figure 2.
L257: The Legend of Figure 2: Figure 2. “Flow curves showing apparent viscosity in functions of shear rate of samples M1 (a), M2 and M6 (b), and shear stress of M1 (c), M2 and M6 (d) of yogurts with addition from hydrocolloids of butternut squash seed as stabilizer.”
5. With reference to Figure 2, are all the measured values reliable in terms of torque? I am just mindful of the fact that the torque of some points at low shear may be smaller than the resolution of the instrument.
RE: The data acquired in rheological assays are reliable. The min toque of rotation values is 0.02 μNm and the maximum 200 mNm; then, all data of torque values were within the range. The rheometer (Modular Advanced Rheometer System Haake Mars 60, Thermo-Scientific, Germany) used.
6. Can the authors clarify the origin of the minimum observed in Figure 4?
RE: In the figure 4, shown the stress sweep that determine the viscoelastic linear region (VLR), evaluating the behavior of the storage (G^') and loss modulus (G^''). In the Line 305, “It was observed that in all cases at low-stress conditions, the modulus G^' and G^'' were linear, and when the stress was sufficiently large, a substantial decrease, showing the slope tends to zero. At this point, the linearity profile deviated; it is considered the critical stress considering as the static yield stress τ_c.”
7. On Page 11, it is written that G’ of M1 is larger than M2. However, from the graph it seems that M1<M2.
RE: The text was improved. The following text was modified:
L341: “Values of G^' at ω = 1 rad·s-1 (Table 7) were used to compare the effect of stabilizers on the samples, presenting a decrease with the addition of HBSS (p < 0.05), where G^' was in the following order M2 ≥ M3 ≥ M5 > M6.”